# Modification of a Shockley-Type Surface State on Pt(111) upon Deposition of Gold Thin Layers

**DOI:** 10.3390/ma11122569

**Published:** 2018-12-17

**Authors:** Igor V. Silkin, Yury M. Koroteev, Vyacheslav M. Silkin, Evgueni V. Chulkov

**Affiliations:** 1Department of Physics, Tomsk State University, Tomsk 634050, Russia; koroteev@ispms.tsc.ru; 2Institute of Strength Physics and Materials Science, Siberian Branch of Russian Academy of Sciences, Tomsk 634050, Russia; 3Department of Physics, Saint Petersburg State University, Saint Petersburg 198504, Russia; 4Department of Material Physics, University of the Basque Country, 20080 San Sebastián/Donostia, Basque Country, Spain; waxslavas@ehu.es (V.M.S.); evguenivladimirovich.tchoulkov@ehu.eus (E.V.C.); 5Donostia International Physics Center (DIPC), 20018 San Sebastián/Donostia, Basque Country, Spain; 6IKERBASQUE, Basque Foundation for Science, 48013 Bilbao, Basque Country, Spain; 7Materials Physics Center MPC, Mixted Centre CSIC–UPV/EHU, 20018 San Sebastián/Donostia, Basque Country, Spain

**Keywords:** platinum, gold, adlayers, density functional calculations, electronic surface states, quantum-well states, spin-orbit splitting

## Abstract

We present a first-principles fully-relativistic study of surface and interface states in the *n* one monolayer (ML) Au/Pt(111) heterostructures. The modification of an unoccupied s−p-type surface state existing on a Pt(111) surface at the surface Brillouin zone center upon deposition of a few atomic Au layers is investigated. In particular, we find that the transformation process of such a surface state upon variation of the Au adlayer thickness crucially depends on the nature of the relevant quantum state in the adsorbate. When the Au adlayer consists of one or two monolayers and this relevant state has energy above the Pt(111) surface state position, the latter shifts downward upon approaching the Au adlayer. As a result, in the 1 ML Au/Pt(111) and 2 ML Au/Pt(111) heterostructures at the equilibrium adlayer position, the Pt-derived surface state experiences strong hybridization with the bulk electronic states and becomes a strong occupied resonance. In contrast, when the number *n* of atomic layers in the Au films increases to three or more, the Pt(111) surface state shifts upward upon reduction of the distance between the Pt(111) surface and the Au adlayer. At equilibrium, the Pt-derived surface state transforms into an unoccupied quantum-well state of the Au adlayer. This change is explained by the fact that the relevant electronic state in free-standing Au films with n≥3 has lower energy in comparison to the Pt(111) surface state.

## 1. Introduction

Many atomically-clean metal surfaces support the electronic states tightly bound to the “metal-vacuum” interface: the surface states [1,2,3] and the image-potential states [4,5,6]. The wave function of the surface states has its maximum in the vicinity of a topmost atomic layer, whereas the image-potential states reside mainly in the vacuum [4,6]. The energetic position (frequently close to the Fermi level) and spatial localization of the surface states ensure its importance in many phenomena occurring at the surface. Moreover, since the wave function of surface states decays into the vacuum significantly slower than the bulk-like electronic states, at certain distances from the surface, the total or partial electron density is determined by a surface state [7]. For this reason, e.g., in scanning tunneling microscopy (STM) experiments, this density is mainly probed. Due to the relevance in many practical applications and from the fundamental point of view, the surface states have been the subject of intense research for decades.

A prototypical example of a surface state is a Shockley s−p surface state residing in a wide energy gap in the center of the surface Brillouin (SBZ) zone of Be(0001) [8,9], noble metal [10,11,12,13,14,15], and transition metal [16,17,18,19,20,21,22,23,24,25] (111) surfaces. In different metals, the energy position of such a state is different with respect to the Fermi level. Thus, on Be(0001), Cu(111), Ag(111), and Au(111), this state has parabolic-like dispersion and is partly occupied with the bottom below the Fermi level at the SBZ center, while on Pd(111) and Pt(111), it is totally unoccupied. There is another difference between these two groups of surfaces. Whereas in Be, Cu, Ag, and Au, the s−p surface states locate inside an energy gap rather far from bulk states and can be traced over a large portion of the SBZ before leaving the energy gap and transforming to the resonance states, in the case of Pd and Pt, this surface state resides so close to the bottom border of the energy gap that sometimes it is difficult to trace its dispersion beyond the SBZ center.

Since the electronic surface states reside at a metal surface, their properties can be modified by the deposition of some amount of adatoms or by adlayers of different kinds [26]. In such a way, the energy and spatial localization of surface states can be greatly varied. Moreover, the adsorbates may introduce their own electronic states, and the final picture often is rather complex. A basic picture of the modification of the electronic surface states of metal surfaces upon deposition of adsorbates was mainly constructed on the basis of the adsorption of prototypical simple metal atoms [27,28,29,30,31,32,33,34,35,36,37,38], noble metals [26,39,40,41,42,43], oxygen [44,45], or hydrogen [46,47]. One can try to interpret the electronic states of a composite adlayer/substrate system as the substrate surface state bands modified by the interaction with the overlayer or consider them as the states of the free-standing adlayer influenced by the presence of the substrate [26]. However, when the coupling between substrate and adsorbing species is strong, the final picture may be rather difficult to understand in simple terms.

In this paper, we consider a strong coupling case studying transformation of the electronic structure of the Pt(111) surface (characterized by an unoccupied surface state just above the Fermi level) upon deposition of a few atomic layers of gold. The catalytic properties of Au/Pt(111) surfaces have been investigated [48,49,50,51] since the 1980s. Recent experimental work demonstrated that when the Au coverage exceeds one monolayer (ML), the growth occurs in a layer-by-layer manner [52]. We find that deposition of the Au adlayers strongly modifies the Pt(111) surface electronic structure around the Fermi level. Moreover, modifications depend on the Au adlayer thickness, and for a sufficiently thick Au adlayer, the resulting electronic structure resembles that of a clean Au(111) surface, where the s−p surface state is partly occupied. In order to unveil the details of the process of the transformation of surface states of the clean Pt(111) surface into those of composite *n* ML Au/Pt(11) systems, we studied the evolution of the electronic structure versus the distance between the Pt(111) surface and the Au adlayers choosing n=1,2,3, and 7. Besides, it helped us in understanding how the electronic states evolve under adding the adlayers, and this may be relevant to the STM experiments showing the scale at which the surface electronic states can be affected by approaching other objects from the vacuum side.

The rest of the paper is organized as follows: In Section 2, we describe the details of the ab initio calculation of the electronic structure of the systems under study. In Section 3, we present the calculated results and discussion. The main conclusions of this work are presented in Section 4.

## 2. Calculation Details

The calculations were carried out within the framework of the electron density functional theory formalism by the projector augmented wave method [53], implemented in the Vienna Ab-initio Simulation Package (VASP) [54,55,56]. For the description of the exchange-correlation effects, the generalized gradient approximation (GGA) in the Perdew–Burke–Ernzerhof parameterization was used [57]. Scalar-relativistic corrections were included in the Hamiltonian, and the spin-orbit interaction was taken into account according to the second variation method [58]. The self-consistency of the electron density was carried out on a (11 × 11 × 1) grid of $k_{∥}$-points in the SBZ constructed by the Monkhorst–Pack scheme [59]. A semi-infinite Pt(111) crystal covered by a Au adlayer consisting of *n* atomic layers was simulated with the *n* ML Au/Pt(111) systems, where a 23-layer film was employed for the description of the platinum substrate and the adlayers were placed on both sides of the slab. A 10 Å vacuum gap between slabs was used. The in-plane lattice parameter for the Au adlayers was chosen to be equal to the Pt bulk constant (3.978 Å) obtained in this work, which is about 2% smaller than the Au lattice parameter [60]. Our structural parameters for the Pt(111) are consistent with the experimental [52,61,62,63,64,65,66,67,68] and calculated [52,69,70,71] data. For each system under study here, we performed optimization of the vertical atomic positions in the Au adsorbate layers and four outer layers of the Pt substrate on each side of the film. The atomic positions of fifteen internal Pt atomic layers were frozen. The interplanar distances were optimized so that the forces acting on the ions did not exceed 0.01 eV/Å.

## 3. Calculation Results and Discussion

### 3.1. 1 ML Au/Pt(111)

When the Au atomic layer was located at a distance $d_{Au-Pt}=$ 10 Å from the Pt surface layer, its electronic structures were essentially decoupled, as can be seen in Figure 1a. The electronic structure of the Au monolayer was characterized by several energy bands in the considered energy window. In Figure 1a, one can see that the Fermi level was intersected by the energy band highlighted by the red thick dashed line, which experienced a strong s−d hybridization. Around the SBZ center (the $\bar{\Gamma }$ point), this band denoted as $S_{3}$ had a rather flat dispersion at energies in the −0.6–−1.0 eV interval, and its orbital composition was mainly of the *d*-character. Its spatial distribution was closely confined to the Au atomic layer, as can be appreciated in the lower panel of Figure 1f, where the charge density of the $S_{3}$ state averaged in the plane parallel to the surface is presented. Beyond the $\bar{\Gamma }$ point vicinity, this band had mainly a *s* character and possessed strong upward dispersion crossing the Fermi level. Note that with increasing the number of atomic layers in the Au slab, the number of such states gradually increased, as well forming in the momentum-energy phase space a region of so-called quantum-well states [39]. Eventually, in the limit of semi-infinite Au crystal, the number of such states became infinite, and they formed a continuum in the projected energy band structure of the Au(111) surface. In Figure 1a in the energy region below −1 eV, one can find several other electronic states characteristic of a free Au monolayer. Since they are not of interest here, we shall not discuss such states anymore. In Figure 1a, above the Fermi level, one can find the energy band with an upward parabolic-like dispersion. Its bottom was located at 2.0 eV and denoted as $S_{1}$. Its localization at the Au monolayer was confirmed by the upper panel of Figure 1f, where the charge density of the $S_{1}$ state as a function of the coordinate *z* perpendicular to the surface is shown.

The $S_{1}$ state did not present any spin splitting. This can be understood considering that the potential anisotropy induced by the Pt surface in the vicinity of the Au atomic layer was small and not sufficient to break appreciably the inversion symmetry. When the $S_{1}$ state approached the upper energy gap border, its localization in the Au-Pt interface was marginal, and the potential asymmetry in this region did not affect the spin degeneracy either.

The electronic structure of the clean Pt(111) surface was characterized by the continuum of the projected bulk electronic states (shown by light blue regions in Figure 1a). In our calculation employing a 23-ML thick Pt(111) slab, this continuum is represented by thin blue solid lines. In this continuum, one can resolve energy bands of two surface states shown by thick blue dashed lines in Figure 1a. The upper surface band has a strong upward dispersion upon moving from the $\bar{\Gamma }$ point close to the lower border of the wide energy gap spanning the energy interval from 0.3–5.6 eV at $\bar{\Gamma }$. Its dispersion was in a good agreement with the experimental data [24] and first-principles calculations [23,24,72,73]. The second surface band had a downward dispersion moving from the SBZ center and became even partly occupied at finite wave vectors in both the $\bar{\Gamma }\bar{M}$ and $\bar{\Gamma }\bar{K}$ symmetry directions of the SBZ. Eventually, both surface bands reached each other at the $\bar{\Gamma }$ point. In Figure 1a, the surface state at this crossing point is denoted as $S_{2}$. Its charge density distribution reported in the middle panel of Figure 1f confirmed its strong confinement to the Pt(111) surface. Note that in the vicinity of the $\bar{\Gamma }$ point, these two surface bands experienced strong hybridization with the bulk-like electronic states, and its dispersion details were extremely sensitive to the calculation methods employed. For instance, in the local-density-approximation (LDA) calculations, the lower energy band dispersion was more flat, and its top was located at about 0.1 eV above the Fermi level [73,74]. Since in this work, we employ the GGA approximation for the description of the exchange-correlation potential, our dispersion of the lower energy Pt(111) surface state was closer to that obtained in [73]. Strong hybridization of these bands with the Pt bulk electronic states [73,74] did not allow the application of the Bychkov–Rashba model [75] in order to quantify the spin splitting. The detailed analysis of other surface states of clean Pt(111) surface can be found in [73,74,76,77,78].

When the distance between the Au atomic layer and the Pt(111) surface layer was reduced to 5 Å the hybridization between the surface states became notable. As seen in Figure 1b, the $S_{1}$ state shifted upward by about 0.1 eV. In its charge density distribution of Figure 1g, one can observe significant penetration of its wave function into several top Pt atomic layers. On the other hand, the $S_{2}$ surface state shifted downward by about 0.03 eV, and as a whole, this energy band shifted downward as well. Interaction with the $S_{1}$ band affected the surface localization of this band at energies above 2.5 eV, resulting in its more efficient scattering into the bulk-like electronic states. Despite the increasing overlap with the continuum of the projected bulk states, the wave function of the $S_{2}$ state maintained its strong localization at the surface. Moreover, it had a non-zero amplitude at the Au monolayer, as seen in Figure 1g. The downward shift of the Shockley surface state in Pt(111) as a result of the interaction with the adlayer s−p electronic states correlated with the general picture established in the simple metal adsorption studies. Since the lower energy states of both the Au monolayer and the Pt substrate were mainly of a *d* character, they were hardly affected upon changing the adlayer-surface distance from 10 Å to 5 Å. At the same time, this was not true for the electronic states close to the vacuum level. In the present case, the vacuum level for the Pt(111) surface was at 6.1 eV above the Fermi level. Comparison of Figure 1a,b reveals strong modifications in the electronic states at energies above ∼5.5 eV. This can be explained by quantization of the vacuum states occurring below the vacuum level in the spatial regions beyond the atomic layers. Such quantized vacuum states resembling the image-potential states [4] appeared in the first-principles calculations due to the intrinsic problem in the description of the correct long-range behavior of the charge density on the vacuum side [79].

In Figure 1c, we show the electronic system of the 1 ML Au/Pt(111) system when the distance between the Au adlayer, and the top Pt layer is put to 4 Å. One can see that the hybridization between the Au and Pt electronic systems significantly increased in comparison with the previously-discussed cases. The $S_{1}$ state in this case had energy of 2.7 eV. As is evident from Figure 1h, its wave function significantly penetrated into the two upper Pt layers. At the same time, one can notice that the wave function of this state had a node in the region between the Au and Pt layers, signaling its anti-bonding character. In contrast, the Pt-derived $S_{2}$ surface state reduced its energy to 0.17 eV accompanied by a general downward shift of the whole energy band. In such a way, the $S_{2}$ surface band interacted more strongly with the bulk-like Pt states, which resulted in the reduction of its dispersion to energies below ∼2 eV. Partly, this scattering is compensated by the transition of a portion of its wave function into the Au layer, as seen in Figure 1h. From Figure 1h, it is clear that this state had a “bonding” character, since its wave function had no node in the region between the Au layer and Pt crystal.

The Au-derived $S_{3}$ state was not influenced significantly by the presence of the substrate at a distance of 4 Å. Only small downward shift of the $S_{3}$ was observed at this distance. Nevertheless, it seems there was some non-negligible hybridization with the bulk electronic structure of Pt at this energy, since in Figure 1h, the barely visible amplitude of $S_{3}$ in the Pt interior can be detected. This is accompanied by changing of the wave function shape in the vicinity of the Au layer.

Reduction of Au-Pt interlayer distance $d_{Au-Pt}$ to 3 Å led to strong modification of the electronic structure of the 1 ML Au/Pt(111) heterostructure, as is obvious from Figure 1d. In such a system, the $S_{1}$ state resided at an energy of 4.3 eV. Its wave function shown in Figure 1i presented strong hybridization between Au and Pt atomic layers. Moreover, its expansion and localization in the vacuum side significantly increased. On the other hand, its penetration into the Pt crystal seemed to be stabilized by the presence of the Pt(111) energy gap. Its wave function had almost zero amplitude inside the Pt crystal beyond the fourth atomic layer. Again, the $S_{1}$ wave function maintained a node in the Au-Pt interlayer space.

More dramatic changes were experienced by the $S_{2}$ surface state. As seen in Figure 1d, at the $\bar{\Gamma }$ point, it became occupied with an energy of −0.07 eV. This value was close to the energy position of a peak seen at −0.2 eV in the photoemission measurements for 1 ML Au/Pt(111) [77]. Moreover, strong hybridization with the electronic states of the substrate strongly reduced the space where the surface bands crossing at the $S_{2}$ point existed. As is evident from Figure 1d, only a small Dirac cone was formed by these surface states. At this point, we should note that determination of the resonance dispersion was difficult and may be rather arbitrary. In this work, we traced the dispersion of resonances based on a careful analysis of the spatial localization of each state involved.

Curiously, the $S_{3}$ state had significantly lower energy when $d_{Au-Pt}=$ 3 Å. In the case of Figure 1d, it located at −1.4 eV at the $\bar{\Gamma }$ point. Nevertheless, its localization in the Au layer was almost not affected by such a significant energy shift. From Figure 1f–i, one can observe that its wave function in Figure 1i was closer to those in Figure 1f,g and differed more from that of the more closely-placed Au layer of Figure 1h.

Figure 1e presents the electronic structure evaluated at the equilibrium interlayer distance $d_{Au-Pt}$ = 2.45 Å between the Au and top Pt atomic layers. The tendencies in the evolution of the surface states noted in the previous cases were observed in this case as well. The $S_{1}$ surface state finally arrived at its equilibrium energy position with an energy of 4.9 eV at the $\bar{\Gamma }$ point. Now, it was located much closer to the upper border of the Pt(111) energy gap than it was in the free-standing Au monolayer case. Its dispersion was almost free-electron-like with effective mass close to unity. The distinct energy position of this surface state respective to the Pt(111) energy gap determined its spatial localization. In Figure 1j, one can observe that now its wave function was almost entirely located on the vacuum side. Due to its energy and spatial localizations, this state should be very sensitive to the details of the potential variations on the vacuum side. As such, it can be considered as an image-potential state of a whole system [80,81]. A similar state was detected in ultra-thin Ag films on Pd(111) [40,41].

At the equilibrium atomic positions in the 1 ML Au/Pt(111) system, the $S_{2}$ surface state can still be unambiguously resolved in the calculated electronic structure. As seen in Figure 1e, it had an energy of −0.1 eV at the $\bar{\Gamma }$ point. Two energy bands crossing at $S_{2}$ formed a Dirac cone and existed in a small wave vector region around the $\bar{\Gamma }$ point. Despite being located in the energy region where the energy gap was not present in the projected electronic structure of Pt(111), the $S_{2}$ surface state was characterized by a strong spatial localization at the surface. Thus, in Figure 1j, its wave function has a maximum above the Au layer, and its amplitude gradually reduced moving into the Pt crystal interior. Inside the crystal beyond the ninth Pt atomic layer from the surface, its wave function can be hardly detected. Its “bonding” character can be confirmed by the lack of a node in the Au-Pt interlayer region, although inside the crystal, it had local minima exactly in the interlayer space.

The $S_{3}$ state had an energy of −1.8 eV in the equilibrium case. Its wave function shown in Figure 1j presented some hybridization between the Au and top Pt atomic layers. Its small portion localized in the vicinity of the top Pt layer. This distribution demonstrated at which scale the strongly-localized *d*-like electronic states participated in the bonding formation of the Au-Pt interface. Moreover, from Figure 1a–e, one can see that two other Pt *d* electronic states located at −1.35 and −1.66 eV in the $\bar{K}$ point vicinity in energy gap below the Fermi level were not altered by adsorption of the Au adlayer at all. This can be explained by the existence of an energy gap in the projected band structure of Au(111) in this region [82,83]. We would like to note that in the LDA calculations [73,74], these and other *d*-like states with energies above −3 eV resided at slightly lower energies, less than 0.2 eV. The maximal difference of 0.6 eV was observed at the bottom of the valence band. At the moment, it is not clear which approximation (LDA or GGA) is better for the description of such states. Since a more detailed analysis of the evolution of such *d*-like states upon interface formation is beyond the scope of this work, we shall not discuss them in the other systems.

### 3.2. 2 ML Au/Pt(111)

In this part, we discuss the surface electronic states formed in the Au/Pt(111) heterostructure when the number of the Au atomic layers increased to two. In Figure 2a, the electronic structure of the 2 ML Au/Pt(111) system with a distance of 10 Å between the nearest Au and Pt atomic layers is reported. One can see that in the almost free-standing 2 ML Au slab, the number of bulk-like quantum-well states increased as well. Now, two such energy bands crossed the Fermi level. As for the $S_{1}$ electronic state, its energy position was notably lower in comparison with the 1 Au ML case. At the $\bar{\Gamma }$ point, now it located at 0.45 eV above the Fermi level, i.e., only 0.17 eV above the $S_{2}$ Pt surface state, which had an energy of 0.28 eV. Since the Au adlayer was effectively decoupled from the Pt surface, the $S_{1}$ band and the upper energy one emerging from the $S_{2}$ point crossed each other without hybridization. The $S_{1}$ charge density distribution reported in Figure 2f confirms its localization in the Au bilayer. The charge density distribution of the $S_{2}$ state also shown in Figure 2f was essentially the same as presented in Figure 1f. Nevertheless, close inspection of Figure 1f and Figure 2f reveals some differences in the spatial decay length into the Pt interior of the $S_{2}$ state in these two cases. It seems the addition of the second Au atomic layer at a distance of about 12 Å from the Pt surface increased the level of confinement of the $S_{2}$ state to the surface region and reduced its penetration depth into the crystal. This is accompanied by the downward energy shift of 0.02 eV.

Approaching the Au bilayer to the Pt surface gradually switched on the interaction between both systems, resulting in hybridization of the electronic states most expanded into the vacuum side. In the case when $d_{Au-Pt}=$ 5 Å the electronic structure is presented in Figure 2b. At this distance, the hybridization between the $S_{1}$ and $S_{2}$ surface states resulted in breaking of the upper energy band emanating from the $S_{2}$ point. This band cannot be unambiguously resolved at energies above ∼0.6 eV. This is accompanied by the downward shift of the $S_{2}$ state, which has an energy of 0.15 eV at the $\bar{\Gamma }$ point. On the contrary, the Au-derived $S_{1}$ state shifted upward and, in the case of Figure 2b, formed the bottom of the energy band linked to the $S_{2}$ point in the case of $d_{Au-Pt}=$ 10 Å. Additionally, close to the bottom border of the Pt energy gap in vicinity of the $S_{1}$ state, we find a second band initially localized in the free-standing Au bilayer. Such strong hybridization between the $S_{1}$ and $S_{2}$ energy bands was reflected in their spatial localization. Figure 2g confirms that in this case, both electronic states had a comparable level of localization in both the Au bilayer and the Pt surface region. Nevertheless, careful analysis of the evolution of these states at the intermediate distances between 10 and 5 Å confirms its origin as assigned in Figure 2b. At any distance, the states $S_{1}$ and $S_{2}$ did not cross each other at the $\bar{\Gamma }$ point, and the avoiding-crossing scenario of the corresponding energy bands was evident.

In Figure 2c, the electronic structure for the 2 ML Au/Pt(111) system with a Au-Pt interlayer distance of 4 Å is shown. The reduction of the distance between the Au bilayer and the Pt surface led to the upward shift of the $S_{1}$ state to 1.55 eV. Figure 2h confirms that at this distance, its wave function was approximately equally distributed between the Au and Pt systems. The $S_{1}$ energy band can be clearly resolved up to an energy of about 3.5 eV, where it lost its surface character upon approaching the Pt energy gap borders. In the nearby region, it experienced strong hybridization, and the second band having its origin in the former $S_{2}$ band of a free staying Au bilayer can be found at upper energies.

A distance of 4 Å between the Au and Pt atomic layers forming the Au-Pt interface provoked a notable downward shift of the Pt-derived surface bands merging at the $S_{2}$ point. The energy at this point was −0.16 eV according to the Fermi level. As a result, the lower band became totally occupied, and the upper energy one resided mainly below the Fermi level. The charge density of the $S_{2}$ state reported in Figure 2h reveals its localization both in the Au bilayer and the Pt surface region. On contrary, one can notice that its penetration into the Pt interior was increased as well. This can be explained by the increasing drift of its energy from the energy gap border towards the interior of the Pt projected bulk band continuum.

When we reduced the distance between the nearby Au and Pt atomic layers to 3 Å, the impact of the hybridization of the electronic states increased on a scale observed in the case of the 1 ML Au/Pt(111) heterostructure. More precisely in Figure 2d. the $S_{1}$ surface state had an energy of 3.5 eV at the $\bar{\Gamma }$ point, i.e., by about 0.8 eV lower than in the 1 ML Au/Pt(111) case. As Figure 2i demonstrates, its wave function was divided in a similar proportion between Au and Pt regions. Again at the energy gap borders, the corresponding energy band experienced strong hybridization with the energy gap borders in both symmetry directions and after leaving this gap upon the increase of the wave vectors became a strong resonance.

The $S_{2}$ state and both surface bands crossing at this point became totally occupied at the interface distance of 3 Å. At the $\bar{\Gamma }$ point, the energy of the $S_{2}$ state was −0.36 eV. As seen in Figure 2, this was accompanied by a reduction of the phase space in the SBZ where the dispersion of these two bands can be clearly resolved. Moreover, in contrast to the previously-discussed examples, the $S_{2}$ state obtained a strong resonating character, interacting with the bulk-like states, forming the Pt projected continuum. This can be observed in its charge density reported in Figure 2i where the non-vanishing amplitude is obvious over a whole Pt interior.

The electronic structure of the 2 ML Au/Pt heterostructure at equilibrium with $d_{Au-Pt}=$ 2.42 Å is presented in Figure 2e. The $S_{1}$ state reached the energy of 4.7 eV above the Fermi level, and its wave function possessed significant localization in the Au bilayer, as Figure 2j shows. Since the energy position of this state was still rather far from the vacuum level position, its spatial penetration into the vacuum was significantly lower in comparison to what occurred for the deposition of a single Au atomic layer discussed previously. On the other hand, one can see in Figure 2j that this state penetrated rather deeply into the Pt crystal, and its finite amplitude can be traced down to the fifth Pt atomic layer. Despite the appreciable spatial expansion of this state both into the vacuum and the Pt bulk, we interpret this state as a first quantum-well state representing the projected bulk electronic states of the Au(111) surface existing in this energy region [80,84]. Regarding the $S_{2}$ surface state, it can still be detected in the electronic structure of Figure 2e. It had an energy of −0.40 eV at the $\bar{\Gamma }$ point, in agreement with the photoemission data for the 2 ML Au/Pt(111) system [77] where a broad peak centered at a energy of ∼−0.3 eV was observed.

In the nearby region, two energy bands forming the Dirac cone at the $S_{2}$ point can be resolved inside the Pt bulk states continuum. Its resonant nature can be deduced from the charge density reported in Figure 2i, where the nonzero amplitude can be seen inside the Pt crystal. Like in the previous case, at equilibrium, the $S_{1}$ and $S_{2}$ wave functions had an anti-bonding and bonding character, respectively.

Analyzing the electronic structures shown in Figure 2a–e, one can trace different evolutions of two Au-derived energy bands crossing the Fermi level in Figure 2a upon contraction of the Au-Pt interface distance. These two energy bands can be classified as quantum well states of a thin Au slab. At all interface distances considered, the lower energy band kept its dispersion along the $\bar{\Gamma }\bar{K}$ direction almost unaltered. This can be explained by the fact that its hybridization with the Pt derived electronic states was impossible due to the presence in the Pt energy gap. In some regions even having dispersion inside the Pt bulk states continuum, its hybridization with Pt electronic states was not efficient in destroying its surface state character. On the contrary, along the $\bar{\Gamma }\bar{M}$ direction, this band strongly interacted with the Pt electronic states and ceased to exist as a well-defined surface state almost at all wave vectors along this symmetry direction in the lattice at equilibrium. The dispersion of the upper energy Au-derived band overlapped with the Pt bulk states continuum at all wave vectors, resulting in strong interaction. As a result, in the equilibrium lattice, this band did not exit. One can notice that upon reduction of the interface distance, the dispersion of this band was hardly modified. Its disappearance was a result of a gradual loss of its localization at the surface region.

### 3.3. 3 ML Au/Pt(111)

The electronic structure of the 3 ML Au/Pt(111) system with a distance of 10 Å between the nearest Au and Pt atomic layers is presented in Figure 3a. In accord with the three atomic layers in the Au slab, the number of the quantum-well states crossing the Fermi level increased to three as well. Regarding the $S_{1}$ and $S_{2}$ electronic states of main our interest, an important difference can be noticed in their mutual positions. Whereas the energy position of the Pt surface state $S_{2}$ was only slightly changed by the addition of a third atomic layer to the Au slab from the vacuum side, the $S_{1}$ Au-derived state had energy of 0.09 eV above the Fermi level, i.e., it resided 0.20 eV below the $S_{2}$ state. Since the $S_{1}$ band and the upper energy band reaching the $S_{2}$ point had similar effective masses, they did not cross each other at any wave vector in the SBZ. On the contrary, the lower energy band emanating from the $S_{2}$ point crossed the $S_{1}$ energy band close to the $\bar{\Gamma }$ point. Figure 3f confirms the spatial confinement of the $S_{1}$ and $S_{2}$ states to the Au and Pt subsystems, respectively.

Our calculations show that when we gradually reduced the distance between the Au slab and the Pt surface, the $S_{1}$ and $S_{2}$ states experienced energy shifts in the opposite directions observed in the two previously-studied systems. However, in contrast with the n=1 and n=2 cases in the 3 ML Au/Pt(111) system, the $S_{1}$ ($S_{2}$) state started to move in a downward (upward) direction. This movement was accompanied by important modifications in the topology of the corresponding energy bands. Figure 3b reveals that when the Au-Pt interface distance was equal to 5 Å the interaction between electronic states in both systems shifted the $S_{1}$ band to an energy of −0.11 at the $\bar{\Gamma }$ point. Moreover, one can find a second energy band emanating from the $S_{1}$ point with downward dispersion, i.e., the Dirac cone was now formed at the $S_{1}$ point.

Regarding the $S_{2}$ band at $d_{Au-Pt}=$ 5 Å it notably shifted upward and now located inside the energy gap. Its energy at the $\bar{\Gamma }$ point was 0.54 eV. Moreover, we detected two spin-resolved energy bands split according to the Bychkov–Rashba mechanism as is shown in insert of Figure 3b. This allows us to determine two parameters defining this spin-orbit splitting, $\gamma_{SO}=0.9$ eV Å, $\Delta k_{∥}=0.013$ Å${}^{-1}$. We believe that these values were very close to those in case of a clean Pt(111) s−p surface state. These values can be contrasted with $\gamma_{SO}=0.45$ eV Å, $\Delta k_{∥}=0.028$ Å${}^{-1}$ and $\gamma_{SO}=1.3$ eV Å, $\Delta k_{∥}=0.075$ Å${}^{-1}$ for clean Au(111) and Ir(111) surfaces, respectively [73,82,85].

The spatial distribution of the $S_{1}$ and $S_{2}$ states at $d_{Au-Pt}=$ 5 Å reported in Figure 3g reveals its strong localization simultaneously in both the Au and Pt regions. Curiously, a large portion of the $S_{1}$ wave function resided in the interlayer region, confirming its bonding character. One can also note that the penetration of the $S_{1}$ state inside the Pt crystal was rather efficient. Nevertheless, the link of both of these states to the Au-derived $S_{1}$ and Pt-derived $S_{2}$ states in the case of decoupled Au and Pt subsystems can be established.

The downward and upward shifts of the $S_{1}$ and $S_{2}$ states upon reduction of the interface distance are confirmed by Figure 3c, where the electronic structure is shown for the distance of 4 Å. Now, the $S_{1}$ state had an energy of −0.36 eV in the SBZ center and gave the origin to two energy bands forming the Dirac cone. In its charge density distribution of Figure 3h, one can observe that it had a resonance character since no vanishing amplitude can be found in the Pt interior. Nevertheless, this state still had strong localization at the Au-vacuum interface. As for the $S_{2}$ state, its wave function had the largest amplitude in the Au-Pt interface with a node in the middle, which confirms its anti-bonding character.

Figure 3d,e demonstrates that at an interface distance of 3 Å and the equilibrium with $d_{Au-Pt}=$ 2.42 Å, the $S_{1}$ state existed at energies of −0.49 and −0.51 eV, respectively. The latter value correlated closely with the presence of a peak at ∼−0.4 eV in the photoemission measurements for the 3 ML Au/Pt(111) system [77]. As Figure 3i,j reveals, the corresponding wave functions had notable resonance components. Nevertheless, these wave functions had strong localization at the Au upmost atomic layer. One can think that it represents a well-known s−p surface state at the clean Au(111) surface. However, at these interface distances in 3 ML Au/Pt(111), we can find only a single double-degenerate band originating from the $S_{1}$ state with no spin-splitting.

As seen in Figure 3d,e, the $S_{2}$ state approached energies of 3.1 and 4.2 eV at a distance of 4 Å and the equilibrium, respectively. This upward shift was accompanied by a gradual reduction of the spin splitting of the corresponding two energy bands. Thus, at the equilibrium, the spin splitting was marginal. Being located significantly lower with respect to the vacuum level, the $S_{2}$ state did not penetrate significantly into the vacuum, as is clear from Figure 3j. Its wave function mainly located in the Au adlayer with the decay length inside the Pt crystal similar to the 2 ML Au/Pt(111) case. This confirms our classification of electronic states in this energy region as the Au-derived quantum-well states.

### 3.4. 7 ML Au/Pt(111)

Upon increasing the thickness of the Au slab, the tendencies observed in the evolution of the electronic states in the 3 ML Au/Pt/(111) system with the variation of $d_{Au-Pt}$ were maintained. The number of Au-derived quantum-well states increased in accord with the number of Au atomic layers in the adlayer and the energy of the $S_{1}$ state sinks as well. As an example, in this part, we present the calculated electronic structure of the 7 ML Au/Pt(111) system. The Au slab was sufficiently thick to gain some knowledge about the electronic structure of the Au(111) surface. In such a way, this system may serve as an example of the evolution of the electronic states upon the approach of two massive bodies with the s−p occupied and unoccupied surface states.

In Figure 4a, the electronic structure of the 7 ML Au/Pt(111) system at $d_{Au-Pt}$= 10 Å is shown. Since the number of the quantum-well states in the Au slab [83] now was relatively large, we no longer highlight them in the figure by thick dashed lines. One can see in Figure 4a that the Au $S_{1}$ band became partly occupied with an energy of −0.51 eV at the $\bar{\Gamma }$ point. Indeed, it represents the antisymmetric combination of the surface states localized at two free surfaces of an Au slab, as can be realized considering the localization of its charge density in Figure 4f with the maxima at both surface atomic layers. A band representing the symmetric combination of the surface states of each Au(111) surfaces exists at lower energies in Figure 4a and had an energy of −0.85 eV at the $\bar{\Gamma }$ point.

Notice that in this situation, the Pt $S_{2}$ surface state had an energy of 0.27 eV at $\bar{\Gamma }$, i.e., confirming that the energy of this surface state can vary at a scale of a few tens of meV by the presence of an external metallic object at a distance typical for STM experiments. Comparing the shape of its charge density in Figure 4f with that for other systems, one can conclude that indeed, the Au film placed at $d_{Au-Pt}$= 10 Å can disturb it rather notably.

When $d_{Au-Pt}$ reduced to 5 Å we observe in the calculated electronic structure of Figure 4b that the $S_{1}$ and $S_{2}$ states changed their energies in a fashion similar to the 3 ML Au/Pt(111) case. Now, the $S_{1}$ state had an energy of −0.54 eV, and its amplitude increased on the outer side of the Au film, as seen in Figure 4g. The finite amplitude of this state in the Pt bulk confirms its resonance character. The downward shift of the $S_{2}$ state resulted in its energy position at 0.42 eV. It maintained its localization at the Pt surface with small penetration into the adjusted surface of the Au slab.

However, the situation was more complex since the symmetric surface state in the Au slab shifted upward with the reduction of $d_{Au-Pt}$. Figure 4c shows that at $d_{Au-Pt}=$ 4 Å, these two Au-induced surface state bands met each other at the $\bar{\Gamma }$ point with an energy of −0.66 eV. The charge density of this state reported in Figure 4h was strongly localized at the Au-vacuum interface with only a small amplitude inside the Pt crystal. A local maximum of this wave function in the interface region can be noted in Figure 4h as well, whereas the localization of the $S_{2}$ at the Au-Pt interface increased.

At $d_{Au-Pt}=$ 3 Å the $S_{1}$ state with an energy of −0.70 eV gave the origin to two energy bands, as seen in Figure 4d. At this distance, the red thick dashed curves represent the spin split energy bands. As for the $S_{2}$ state, it increased its energy up to 2.9 eV and, as Figure 4i shows, penetrated significantly into the Au slab.

At equilibrium atomic positions with $d_{Au-Pt}=$ 2.42 Å, the $S_{1}$ spin-spit surface bands can be traced in the energy interval from −0.68 eV to about 2 eV, as depicted in Figure 4e by red thick dashed lines. Its wave function shown in Figure 4j confirms it as a Au(111) surface state with a small resonance admixture since its penetration exceeded the 7 ML Au adlayer thickness. In this system, the $S_{2}$ state had an energy of 3.6 eV, i.e., it located close to the middle of the projected gap of the Pt(111) surface. As a result, it penetrated weakly into the Pt substrate, as seen in Figure 4i. Instead, this state was almost entirely confined to the Au film and can be classified as a Au quantum-well state. A second Au quantum-well state can be observed in Figure 4e at an energy of 4.9 eV.

## 4. Conclusions

We have studied the mechanism of the transformation of electronic surface states upon adsorption of thin metal layers considering as an example *n* ML Au/Pt(111) heterostructures. Following the evolution of relevant surface electronic states upon variation of the distance between the substrate and the adlayers, two different mechanisms of transformation have been found. In particular, the unoccupied s−p surface state of a clean Pt(111) surface became partly occupied when one or two atomic layers of Au were deposited. In such a way, this state became a strong resonance state of the whole system with localization at the surface. The lowest unoccupied s−p state existing in the Au slab in the SBZ center shifted upward in energy upon approaching the Pt(111) surface and became a state with a strong localization in the vacuum at the equilibrium interface distance at n=1. In the case of n=2, it became an Au quantum-well state.

The situation is different when the thickness of the Au slab is three or more atomic layers. In this case, the energy of the relevant quantum state of the free-standing Au slab was lower than the energy of the Pt(111) surface state. This situation results in the upward shift of the energy of the Pt(111) surface state upon approximation of the Au slab. In the equilibrium atomic position, the former Pt(111) surface state transformed into a quantum-well state of the Au slab and increased its energy from 0.30 eV on the clean Pt(111) surface to 4.2 eV and 3.6 eV in the 3 ML Au/Pt(111) and 7 ML Au/Pt(111) heterostructures, respectively. In contrast, the surface state energy of the Au slab with three or more atomic layers experienced a downward shift upon approaching the Pt(111) and transformed into a partly-occupied surface state of the combined system. Eventually, upon the increase of the Au adlayer thickness, such a state evolved into a s−p surface state of a clean Au(111) surface.

The predicted effects can be probed by the angle-resolved photoemission, inverse two-photon photoemission, and time-resolved photoemission. We propose that our findings in the lattices out of equilibrium can be realized in the STM experiments or by the growth of ultra-thin dielectric layers separating the Pt(111) and the Au adlayers.

## Figures and Tables

**Figure 1 materials-11-02569-f001:**
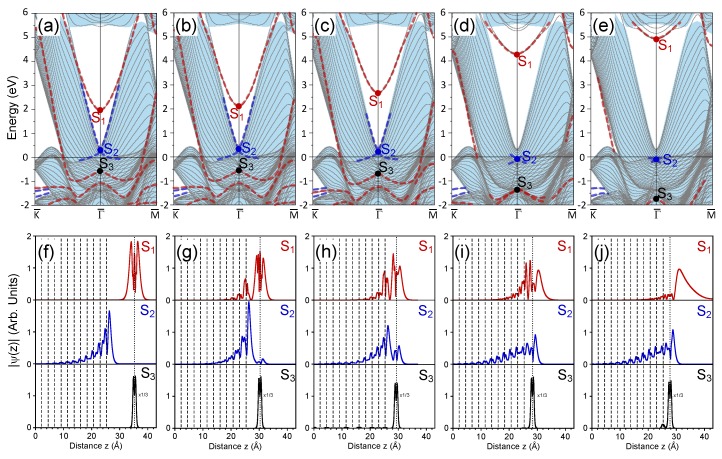
Electronic structure of the one monolayer (ML) Au/Pt(111) surface at (**a**) 10 Å, (**b**) 5 Å, (**c**) 4 Å, (**d**) 3 Å, and (**e**) the equilibrium distance between the top Pt and Au atomic layers. The surface states discussed in the text are marked as $S_{1}$, $S_{2}$, and $S_{3}$. Panels (**f**–**j**) present respectively the in-plane averaged charge density distributions of the $S_{1}$ (red), $S_{2}$ (blue), and $S_{3}$ (black) surface states. In (**f**–**j**), vertical dashed (dotted) lines show the positions of the Pt (Au) atomic layers. The central atomic layer of the 23 ML Pt slab is placed at zero.

**Figure 2 materials-11-02569-f002:**
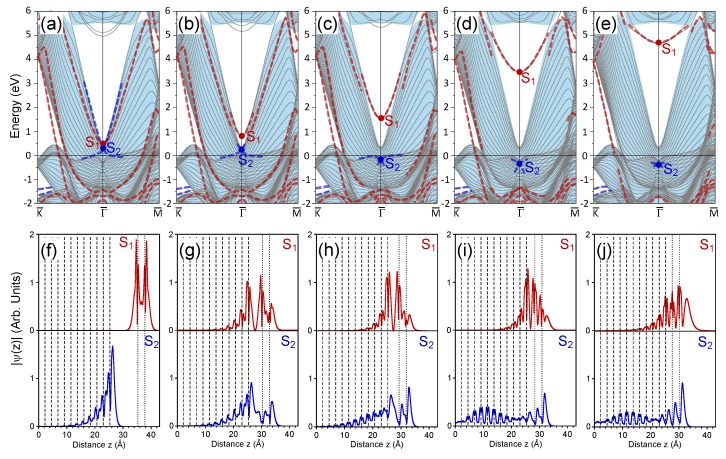
Electronic structure of the 2 ML Au/Pt(111) surface at (**a**) 10 Å, (**b**) 5 Å, (**c**) 4 Å, (**d**) 3 Å, and (**e**) the equilibrium distance between the top Pt and closest Au atomic layers. The surface states discussed in the text are marked as $S_{1}$ and $S_{2}$. Panels (**f**–**j**) present respectively the in-plane averaged charge density distributions of the $S_{1}$ (red) and $S_{2}$ (blue) surface states. In (**f**–**j**), vertical dashed (dotted) lines show the positions of the Pt (Au) atomic layers. The central atomic layer of the 23 ML Pt slab is placed at zero.

**Figure 3 materials-11-02569-f003:**
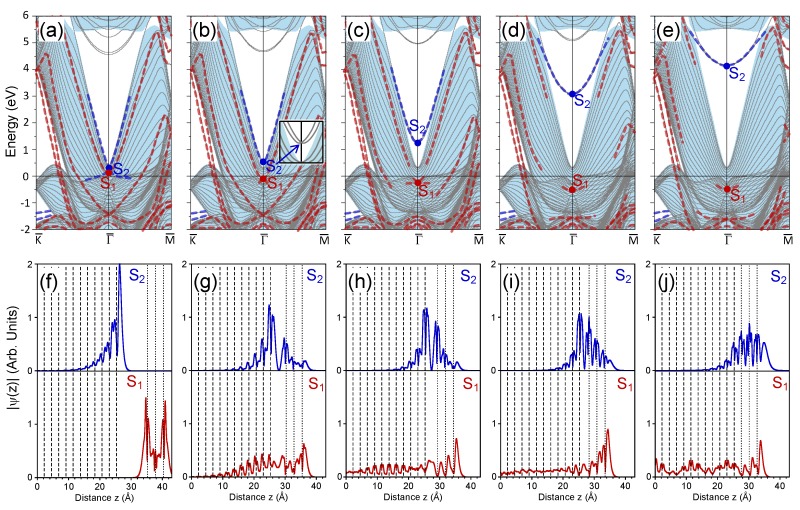
Electronic structure of the 3 ML Au/Pt(111) surface at (**a**) 10 Å, (**b**) 5 Å, (**c**) 4 Å, (**d**) 3 Å, and (**e**) the equilibrium distance between the top Pt and closest Au atomic layers. The surface states discussed in the text are marked as $S_{1}$ and $S_{2}$. In (**b**), insert zooms into the region in the vicinity of the $S_{2}$ state. Panels (**f**–**j**) present respectively the in-plane averaged charge density distributions of the $S_{1}$ (red) and $S_{2}$ (blue) surface states. In (**f**–**j**), vertical dashed (dotted) lines show the positions of the Pt (Au) atomic layers. The central atomic layer of the 23 ML Pt slab is placed at zero.

**Figure 4 materials-11-02569-f004:**
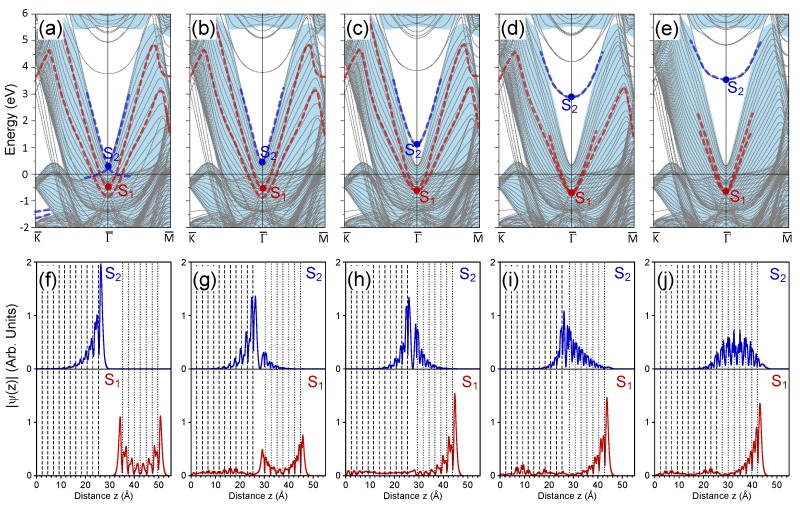
Electronic structure of the 7 ML Au/Pt(111) surface at (**a**) 10 Å, (**b**) 5 Å, (**c**) 4 Å, (**d**) 3 Å, and (**e**) the equilibrium distance between the top Pt and closest Au atomic layers. The surface states discussed in the text are marked as $S_{1}$ and $S_{2}$. Panels (**f**–**j**) present respectively the in-plane averaged charge density distributions of the $S_{1}$ (red) and $S_{2}$ (blue) surface states. In (**f**–**j**), vertical dashed (dotted) lines show the positions of the Pt (Au) atomic layers. The central atomic layer of the 23 ML Pt slab is placed at zero.

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
