# Peer review of "Modification of a Shockley-Type Surface State on Pt(111) upon Deposition of Gold Thin Layers"

_materials, 2018, doi:10.3390/ma11122569_

Round 1
Reviewer 1 Report
In the manuscript, it studied the mechanism of transformation of electronic surface states upon adsorption of thin metal layers considering as an example ML Au/Pt(111) heterostructures. The DFT models were constructed and presented in a scientific manner. The only concern is that there is no direct experimental observation to support the model predictions. It should be more meaningful if the results can be used to understand some existing experimental data. Right now, it is purely computational analysis.
Author Response
Response to Reviewer 1 Comments
Point: In the manuscript, it studied the mechanism of transformation of electronic surface states upon adsorption of thin metal layers considering as an example ML Au/Pt(111) heterostructures. The DFT models were constructed and presented in a scientific manner. The only concern is that there is no direct experimental observation to support the model predictions. It should be more meaningful if the results can be used to understand some existing experimental data. Right now, it is purely computational analysis.
Response: We thank the referee for the positive evaluation of our work. Regarding the comparison with the experiment in the revised version we perform direct comparison of the calculated surface state energies obtained at the equilibrium atomic positions with existing photoemission data for the 1, 2, and 3 monolayers of Au deposited on Pt(111). On the other hand, we believe that this work will attract an experimental interest to such systems and our findings may find experimental support. In the concluding part we refer to some kinds of experiments they can be studied.
Reviewer 2 Report
The authors investigated the surface and interface states in the n ML Au/Pt(111) heterostructures. They found that the transformation of the surface states with respect to the Au-Pt distance is drstically changed when the number of Au adlayers changes. When the number of Au adlayers is one or two, Pt-derived surface states go downward and Au-derived states go upward in energy by decreasing the Au-Pt distance. On the other hand, the situation is inverted when the number of Au adlayers is larger.
Theoretical analysis is conducted very carefully, and their interesting results are well supported by their calculations. I have only minor concerns as follows. I recommend publication if the following points are resolved.
(1) I do not understand the way how the authors resolve energy bands of the surface states shown with thick blue and red dashed lines in Figures. For example, the authors claim "(page 6, line 180-182) Moreover, strong hybridization with the electronic states of the substrate strongly reduces the space where the surface bands crossing at the S2 point exist. As is evident from Fig.1(d) only a small Dirac cone is formed by these surface states", but it is unclear whether this claim is well supported by calculations because the criteria to depict blue dashed lines in Fig.1(d) is not shown.
(2) It might be helpful to show the equilibrium Au-Pt interface distance for all the cases.
(3) minor typos:
[page 3, line 71] projection plane wave method -> projector augmented wave method
[page 7, line 220] On -> One
Author Response
Response to Reviewer 2 Comments
The authors investigated the surface and interface states in the n ML Au/Pt(111) heterostructures. They found that the transformation of the surface states with respect to the Au-Pt distance is drstically changed when the number of Au adlayers changes. When the number of Au adlayers is one or two, Pt-derived surface states go downward and Au-derived states go upward in energy by decreasing the Au-Pt distance. On the other hand, the situation is inverted when the number of Au adlayers is larger.
Theoretical analysis is conducted very carefully, and their interesting results are well supported by their calculations. I have only minor concerns as follows. I recommend publication if the following points are resolved.
Response: We thank the referee for the positive evaluation of our work.
Point 1: I do not understand the way how the authors resolve energy bands of the surface states shown with thick blue and red dashed lines in Figures. For example, the authors claim "(page 6, line 180-182) Moreover, strong hybridization with the electronic states of the substrate strongly reduces the space where the surface bands crossing at the S2 point exist. As is evident from Fig.1(d) only a small Dirac cone is formed by these surface states", but it is unclear whether this claim is well supported by calculations because the criteria to depict blue dashed lines in Fig.1(d) is not shown.
Response 1: We would like to note that the exact determination of the resonance dispersion is difficult and present some ambiguity. Therefore the frequently used determination criteria sometimes is not efficient. This is exactly what occurs in the studied systems. Therefore we performed a much more laborious approach by checking all the states manually. Of course, it also presents some level of uncertainty, but based on our experience we are convinced that it is the best way to determine the dispersion of the resonances in complex situations.
Point 2: It might be helpful to show the equilibrium Au-Pt interface distance for all the cases.
Response 2: In the revised version we included the equilibrium Au-Pt interface distances for all the cases.
Point 3: minor typos:
[page 3, line 71] projection plane wave method -> projector augmented wave method
[page 7, line 220] On -> One
Response 3: These typos and others are corrected.

Reviewer 3 Report
In this work, the authors analyze the evolution of the surface states, most notably their position, hybridization and occupancy, of the Au/Pt(111) system with change of the number of Au layers. This is an important problem and results can be used in the variety of applications. The results are well presented and will be of interest to the audience of Materials.
I recommend the paper for publication after the authors elaborate the following. In the discussion of the results of Fig.1, they mention that the dispersion of the surface states is sensitive to the choice of the XC potential (LDA or GGA). Clearly, in the case of surfaces, the charge gradient is important, but is the PBE correction to LDA accurate enough for the (rather localized) d states? In other words, it would be helpful to add a few words, why PBE is sufficient approximation for this system.
Author Response
Response to Reviewer 3 Comments
Point: In this work, the authors analyze the evolution of the surface states, most notably their position, hybridization and occupancy, of the Au/Pt(111) system with change of the number of Au layers. This is an important problem and results can be used in the variety of applications. The results are well presented and will be of interest to the audience of Materials.
Response: We thank the referee for the positive evaluation of our work.
Point 1: I recommend the paper for publication after the authors elaborate the following. In the discussion of the results of Fig.1, they mention that the dispersion of the surface states is sensitive to the choice of the XC potential (LDA or GGA). Clearly, in the case of surfaces, the charge gradient is important, but is the PBE correction to LDA accurate enough for the (rather localized) d states? In other words, it would be helpful to add a few words, why PBE is sufficient approximation for this system.
Response 1: Comparing the LDA and GGA band structures obtained in Refs. 73, 74 and here one can conclude that the difference in the energies of the d-like surface states above -3 eV is smaller 0.2 eV between two approximations. This difference increases with the increasing binding energies and reaches a maximal value of 0.6 eV at the bottom of the valence band. At the moment it is not clear which approximation (LDA or GGA) is better for the description of such states. We add the corresponding comment into the manuscript.